# From Weather to Weathering: Foundation Models for Carbon Removal via Rock Weathering

**Sasankh Munukutla**[*]
Terradot Soil, Inc.
San Francisco, CA, USA
sasankh@terradot.earth

**Felipe Oviedo**
Microsoft AI for Good
Redmond, WA, USA

**Kenji Takeda**
Microsoft Research
Cambridge, UK

## Abstract

Enhanced Rock Weathering (ERW) harnesses natural silicate weathering, accelerating it by applying crushed rock powder onto soils. ERW has the potential to remove gigatons of carbon dioxide per year, making it a promising solution to stabilize Earth's climate. One of ERW's main challenges to being deployed at scale is cost-effective, rigorous, and accessible Measurement, Reporting, and Verification (MRV), which today requires dense ground-truth sampling and extensive laboratory analysis. We present TerraNova, to our knowledge the first deep-learning-based model that leverages latent weather representations learned by a Foundation Model (FM), Aurora, to predict Carbon Dioxide Removal (CDR) rates of rock weathering in real-world field conditions. Trained on over 170,000 chronosequences of largely natural weathering data and limited ERW data, TerraNova achieves $R^2 = 0.8261$ and MAE $= 0.0750$ on held-out validation sites, demonstrating strong performance across diverse conditions. Through interpretability analysis on synthetic ERW scenarios, we demonstrate alignment with existing geochemical understanding across several key weathering drivers, e.g., higher soil moisture leads to higher CDR rates, showing that TerraNova learns the underlying drivers from largely natural weathering. Our results show the promise of AI and FMs to predict carbon removal and the potential to aid ERW MRV at climate-relevant scales.

## 1 Introduction

The mitigation of climate change requires dramatic reductions in emissions and active Carbon Dioxide Removal (CDR) of ∼10 gigatons from the atmosphere annually by 2050 (IPCC, 2022). Enhanced Rock Weathering (ERW) is a promising CDR pathway that has the potential to remove gigatons of carbon dioxide per year (Strefler et al., 2018). ERW harnesses the natural process by which silicate minerals react with atmospheric $CO_2$ dissolved in rainwater to form stable bicarbonates (Beerling et al., 2020). When finely crushed silicate rocks, such as basalt, are applied to agricultural soils, this weathering process is accelerated by orders of magnitude, removing carbon while having the potential to improve soil health and crop yields (Beerling et al., 2024).

Today, to measure and quantify carbon removal from ERW, dense ground-truth samples are required, including soil samples, soil aqueous samples, and crop samples along with extensive laboratory analysis of the samples to help quantify the CDR (Isometric, 2025). The current approach to using computational models for ERW relies on process-based models such as Reactive Transport Models (RTMs), which require explicitly defining and approximating all relevant processes such as mineral kinetics and soil biogeochemical processes (Beerling et al., 2020). These approaches are extremely helpful for building process understanding and have the potential to perform well at prediction, but can be computationally expensive and require detailed parameterization and calibration (Steefel et al., 2015).

---

[*]Corresponding author.

At the same time, deep learning approaches are increasingly used for a variety of climate and Earth system problems. From weather prediction to flood forecasting, deep learning models match and even outperform numerical/process-based modeling while being computationally more efficient (Bi et al., 2023; Nearing et al., 2024). We investigate the application of deep learning to rock weathering and specifically apply the Foundation Model (FM) Aurora (Bodnar et al., 2024). In particular, weather and Earth system FMs like Aurora have demonstrated great potential at modeling a variety of Earth system processes; here, we are interested in exploring their generalization potential to geochemical processes beyond weather.

While the growing energy demands of AI raise legitimate environmental concerns (You et al., 2025; Bashir et al., 2024), it is compelling to explore how AI and FMs can directly help address climate change.

Our contributions are as follows:

- We introduce a chronosequence-based dataset construction methodology that transforms sparse geochemical measurements into supervised learning examples for CDR rate prediction.
- We systematically evaluate multiple modeling approaches: AutoML baselines, LSTMs, and Neural Network decoders with and without FMs.
- We introduce TerraNova, our best model that achieves $R^2 = 0.8261$ and MAE $= 0.0750$ for CDR rate prediction on held-out validation sites.
- We demonstrate that Aurora's learned latent atmospheric and surface representations substantially outperform raw ERA5 climate data as inputs for CDR rate prediction.
- We perform interpretability analysis showing that the model aligns with known geochemical drivers of weathering.
- Together, these contributions show the potential of FMs and deep learning to aid ERW MRV, which we demonstrate with operational examples.
- More broadly, we demonstrate the potential of weather FMs to generalize beyond atmospheric prediction to geochemical processes, suggesting a pathway for applying FMs to other Earth system applications.

## 2 RELATED WORK

### 2.1 PROCESS-BASED MODELS

The geochemistry community has developed sophisticated process-based models to simulate mineral dissolution and weathering dynamics. These models are grounded in transition state theory (TST), which describes dissolution rates as a function of the distance from thermodynamic equilibrium (Lasaga, 1984; Aagaard & Helgeson, 1982). These kinetic rate laws are then coupled with advection-dispersion equations to simulate weathering in porous media. In soils, weathering is further driven by soil and crop processes such as root and microbial respiration, the release of organic acids (e.g., oxalic, citric) and strong mineral acids through nitrification, all of which can impact weathering and CDR (Beerling et al., 2020).

Modern reactive transport frameworks like CrunchFlow (Steefel et al., 2015), PFLOTRAN (Lichtner et al., 2015), and PHREEQC (Parkhurst & Appelo, 2013) can integrate these soil-plant-mineral interactions (Tang et al., 2016), although doing so requires extensive parameterization and site-specific calibration. These models have then been applied to ERW scenarios to predict dissolution rates, base cation release, and CDR (Kelland et al., 2020).

### 2.2 MACHINE LEARNING FOR WEATHERING DYNAMICS

Prior work on data-driven weathering prediction has largely focused on empirical rate equations fit to laboratory or small-scale field data (Palandri & Kharaka, 2004; Brantley, 2008). Recent efforts have applied machine learning to predict weathering at larger scales. Bush et al. (2026) used random forests to predict silicate weathering rates from climate, lithology, hydrology, and biotic features.

However, these approaches typically require substantial labeled data from geochemical surveys, limiting their applicability to data-sparse settings and solely focus on natural weathering.

Our approach differs by leveraging representations learned by FMs pretrained on abundant weather data, potentially enabling accurate prediction with less task-specific supervision.

## 2.3 EARTH SYSTEM FOUNDATION MODELS

Recent years have witnessed remarkable progress in applying deep learning to weather and climate prediction. Foundation models trained on decades of reanalysis data have achieved state-of-the-art performance across multiple forecasting tasks.

**Aurora** (Bodnar et al., 2024) is a flexible FM for Earth system prediction trained on over a million hours of diverse Earth system data. Aurora uses a 3D Swin Transformer architecture with 3D Perceiver-based encoders and decoders operating on latitude-longitude grids with pressure levels, learning rich spatiotemporal representations that transfer across atmospheric, ocean, and surface prediction tasks. Notably, Aurora achieved state-of-the-art results on air quality forecasting despite not being explicitly trained on chemical transport, suggesting its representations capture generalizable physical processes. Lehmann et al. (2025) adapt Aurora by adding custom decoders to predict unseen physical processes, namely hydrological variables like soil moisture or potential evaporation. This shows the potential of FMs in Earth systems to extend to new variables, which inspires this work.

Pangu-Weather (Bi et al., 2023) and GraphCast (Lam et al., 2023) similarly demonstrate that large-scale pretraining on atmospheric data yields models that outperform numerical weather prediction on many tasks. These successes raise the question of whether such representations can transfer to adjacent Earth system processes like silicate weathering.

## 3 METHODS

### 3.1 DATA

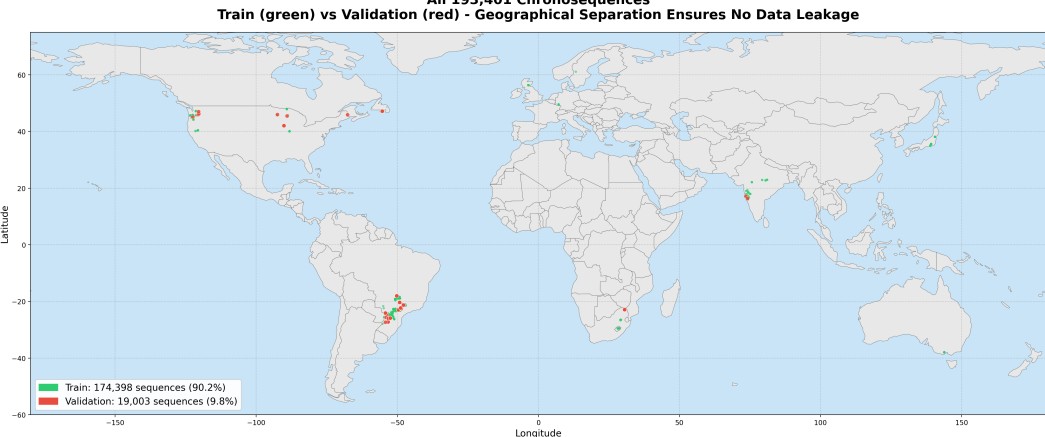

Figure 1: Distribution of train (green) and validation (red) data

We assemble a dataset of natural weathering and ERW chronosequences, aiming to train a model that can learn underlying weathering dynamics across spatial scales and Reactive Surface Area (RSA) values. While both natural weathering and ERW are governed by the same fundamental silicate dissolution chemistry, they differ in important respects: ERW uses finely crushed rock with significantly higher RSA, is applied to actively managed agricultural soils, and operates at field/plot scales rather than the catchment/regional scales typical of natural weathering. These distinct conditions may introduce different controlling variables. Nevertheless, the shared geochemical foundations

combined with the current scarcity of ERW data relative to natural weathering motivate training on a combined dataset to learn general weathering dynamics.

Our feature variables include atmospheric/weather data, namely ERA5 data encoded into Aurora latent representations, soil properties, dynamic environmental properties, amendment properties, and spatial and temporal metadata including temporal duration, area, and location. Our target variable is **Carbon Dioxide Removal (CDR) rate** in tons $CO_2$e per hectare per year (t $CO_2$e/ha/yr).

Natural weathering data are sourced from basaltic catchments worldwide in the GloRiCh (Hartmann et al., 2019) database and in Brazil from ANA (Agência Nacional de Águas e Saneamento Básico (ANA)) and IAET (Instituto Água e Terra, n.d.) databases. We focus on basaltic catchments because ERW today predominantly uses basalt feedstock; extending to other lithologies is left for future work. For natural weathering, CDR rates are derived from stream bicarbonate concentrations, temperature, and runoff using the empirical formulation of Li et al. (2016) that builds upon Dessert et al. (2003). Enhanced Rock Weathering data is sourced from Beerling et al. (2024) and Kantola et al. (2023) field trials. When feature variables are not available in the original studies, we obtain them from external datasets (see Appendix for a complete list of Feature Variables and their sources).

To construct chronosequences, we pair measurements at the same location across different times: for measurements at times $t_i$ and $t_{i+1}$ (where $t_{i+1} > t_i$), we create a training sample $\langle \mathbf{x}_{t_i}, \mathbf{x}_{t_{i+1}} \rangle$ with target $\mathbf{y}_{t_{i+1}}$. We allow the temporal gap $\Delta t = t_{i+1} - t_i$ to vary, enabling the model to learn weathering dynamics across multiple temporal scales; the training data contains gaps ranging from several months to years. For a location with $n$ measurements, this yields $\binom{n}{2}$ distinct chronosequences. We ensure no control-treatment overlap in ERW sequences, i.e., a measurement at a control site is never paired with a measurement at a treatment site.

Using this approach, we assemble a dataset of 193,401 chronosequences from both natural weathering and ERW data (Figure 1). The dataset is dominated by natural weathering chronosequences, reflecting the current scarcity of published ERW field trials.

We split the sequences into training (174,398 / $\sim$90% of sequences, of which there are 700 ERW sequences) and held-out validation (19,003 / $\sim$10% of sequences, of which there are 190 ERW sequences) sets while ensuring no location appears in both sets and that ERW sequences are present in both, allowing us to evaluate generalization to unseen sites.

## 3.2 PROBLEM FORMULATION

We formulate the task as a foundation model transfer task: given pretrained atmospheric representations and site-specific covariates, predict CDR rates. Formally, given a chronosequence at a single site with measurements at times $t_i$ and $t_{i+1}$, we aim to predict CDR rate at $t_{i+1}$.

**Input Representation:** For each chronosequence, the input consists of:

- **Atmospheric/Climate Features, $\mathbf{c} \in \mathbb{R}^{d_c}$, $d_c = 192$:** Aurora latent embeddings extracted from ERA5 reanalysis data spanning $[t_i, t_{i+1}]$, capturing atmospheric conditions at the site.
- **Static Soil Properties, $\mathbf{s} \in \mathbb{R}^{d_s}$, $d_s = 3$:** Soil texture features (sand, silt, clay fractions).
- **Dynamic Site Properties, $\mathbf{d} \in \mathbb{R}^{d_d}$, $d_d = 11$:** Time-varying environmental properties including soil state (pH, CEC, SOC), hydrology (precipitation, moisture, runoff, ET), and vegetation activity (NPP).
- **Amendment Properties, $\mathbf{a} \in \mathbb{R}^{d_a}$, $d_a = 17$:** Amendment properties like mineralogy, RSA, and application rate.
- **Spatial Metadata, $\boldsymbol{p} \in \mathbb{R}^{d_p}$, $d_p = 3$:** Spatial features like latitude, longitude, and area.
- **Temporal Metadata, $\boldsymbol{t} \in \mathbb{R}^{d_t}$, $d_t = 6$:** Temporal features like interval duration $\Delta t$, cyclical month and day-of-year encodings, and normalized year.

All inputs are fixed-length vectors that are then independently encoded (see Figure 2).

**Prediction Target:** The model predicts CDR rate at $t_{i+1}$:

$$\hat{\mathbf{y}}_{t_{i+1}} = f(\mathbf{c}, \mathbf{s}, \mathbf{d}, \mathbf{a}, \boldsymbol{p}, \boldsymbol{t}) = \hat{y}_{t_{i+1}}^{\text{CDR}} \tag{1}$$

where $y^{\text{CDR}}$ is the CDR rate (t $CO_2$e/ha/yr).

**Training Objective:** We train using the standard Mean Squared Error (MSE) as our objective function:

$$\mathcal{L} = \frac{1}{N} \sum_{n=1}^{N} (\hat{y}_n^{\text{CDR}} - y_n^{\text{CDR}})^2 \tag{2}$$

We report Mean Absolute Error (MAE) for evaluation, as it is directly interpretable in physical units (t $CO_2$e/ha/yr).

**Inference:** Given site characteristics and a specified time window, the model predicts CDR rates. By varying the temporal encoding, the same model can provide predictions at different time horizons. Rolling out predictions across intervals $\{[t_0, t_1], [t_1, t_2], \ldots\}$ and summing yields cumulative CDR removal: $\text{CDR}_{\text{total}} = \sum_k \hat{y}_k^{\text{CDR}} \cdot \Delta t_k$.

## 3.3 MODEL ARCHITECTURE

### 3.3.1 TERRANOVA

Figure 2: TerraNova Architecture

TerraNova combines pretrained weather representations from Aurora with site-specific features through a modular encoder-decoder architecture, thereby leveraging an FM with a science-first approach. The model consists of five key components (Figure 2):

**FM Encoder:** We extract latent representations from Aurora's frozen pretrained encoder and backbone. Aurora processes ERA5 inputs comprising surface variables (2m temperature, 10m winds, mean sea level pressure), atmospheric variables (temperature, winds, humidity, geopotential at multiple pressure levels), and static fields (land-sea mask, soil type, orography). The encoder produces spatial feature maps at $0.25°$ resolution.

**Multi-Scale Feature Extraction:** Rather than extracting features at a single point, we capture weathering-relevant processes operating at different spatial scales. For each location, given its field area $A$, we extract features at three scales: (1) *local* (field-level, adaptive to $A$), (2) *regional* ($10 \times A$ context), and (3) *global* (10 km $\times$ 10 km climate context). Each scale uses adaptive average pooling to a fixed $4 \times 4$ spatial grid over 4 latent dimensions (reduced from Aurora's 1024-dim backbone representation using PCA), yielding 64-dimensional embeddings per scale (192 dimensions total).

**Encoders:** Three separate MLPs encode domain-specific site characteristics: a *static soil encoder* for texture (i.e., sand, silt, clay fractions), a *dynamic site encoder* for time-varying environmental properties (i.e., pH, CEC, precipitation, moisture, runoff, evapotranspiration, SOC, erosion, porosity, base saturation, NPP), and an *amendment encoder* for feedstock properties (e.g., oxide composition, mineral content, RSA, application rate). Each encoder maps its inputs through a two-layer MLP with 512 hidden units, GELU activation, dropout (0.1), and layer normalization, producing 64-dimensional embeddings (192 total features).

**Spatial and Temporal Encoders:** A spatial encoder processes normalized coordinates (latitude scaled to $[-1, 1]$, longitude normalized to $[-1, 1]$) and log-transformed field area through a two-layer MLP with GELU activation and layer normalization (64 features). A temporal encoder processes the log-scaled prediction interval $\tau = t_{i+1} - t_i$ along with cyclical calendar features and normalized year at $t_{i+1}$, through a two-layer MLP with GELU activation (64 features).

**Prediction Head:** All encoded features are concatenated (512 dimensions: 192 Aurora + 64 static soil + 64 dynamic site + 64 amendment + 64 spatial + 64 temporal) and passed to a three-layer MLP decoder for CDR prediction. The decoder employs a progressive bottleneck architecture: $512 \rightarrow 512$ (GELU, dropout), $512 \rightarrow 256$ (GELU, dropout), $256 \rightarrow 1$.

### 3.3.2 BASELINES

We compare TerraNova against architectures that systematically ablate the foundation model contribution:

**FLAML (AutoML):** An automated machine learning baseline using FLAML (Wang et al., 2021) trained on chronosequences with tabular site features only (soil, dynamic, amendment, spatial, temporal), without any weather representations.

**PHREEQC:** A geochemical baseline using the PHREEQC reactive transport framework based on (Kelland et al., 2020) with transition state theory kinetics and rate constants calibrated on the training set, representing the process-based modeling approach. We are only able to perform limited calibration of this model.

**LSTM:** Following the success of LSTM architectures for global flood forecasting (Nearing et al., 2024), we train an LSTM encoder-decoder that processes the same input features as TerraNova through an LSTM encoder, with the hidden state transferred to a decoder LSTM conditioned on temporal encodings. We evaluate two variants: *LSTM-ERA5* using raw ERA5 variables directly, and *LSTM-Aurora* using Aurora embeddings as input.

**TerraNova-ERA5:** The full TerraNova architecture with raw ERA5 variables replacing Aurora embeddings, isolating the contribution of FM pretraining from the architectural design.

## 3.4 TRAINING & EVALUATION

We train all models using the AdamW optimizer (Loshchilov & Hutter, 2019) with learning rate $10^{-4}$, weight decay $10^{-2}$, batch size 16 for TerraNova models and batch size 32 for LSTM models. We use a ReduceLROnPlateau learning rate scheduler with factor 0.5 and patience 5. Training proceeds for up to 50 epochs with early stopping (patience 10) based on validation loss. All deep learning model experiments are run on H100 GPUs.

To enable efficient experimentation, we precompute and cache both Aurora embeddings and raw ERA5 variables for all site-date combinations. This reduces per-epoch training time from several hours (with online feature extraction) to minutes, enabling rapid iteration over architectural choices and hyperparameters.

For evaluating model performance, we consider MAE and the Coefficient of Determination $R^2$ as metrics.

Table 1: Validation performance of all experiments

| Experiment | CDR Rate $R^2$ | CDR Rate MAE |
|---|---|---|
| **TerraNova** | **0.8261** | **0.0750** |
| TerraNova-ERA5 | 0.3434 | 0.1454 |
| LSTM-Aurora | 0.6131 | 0.1225 |
| LSTM-ERA5 | 0.2660 | 0.1608 |
| PHREEQC (limited calibration) | 0.0320 | 0.3730 |
| FLAML (XGBoost) | 0.5792 | 0.0914 |

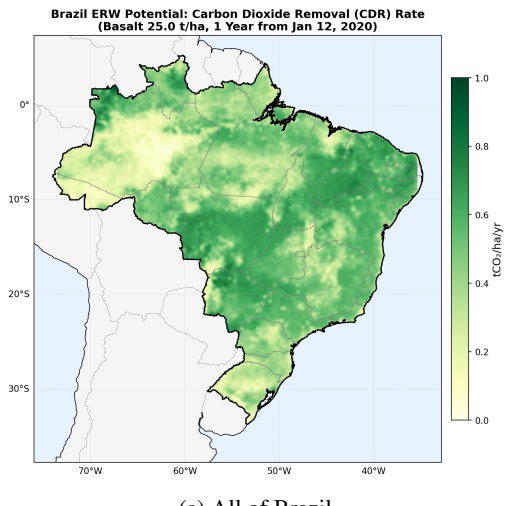

(a) All of Brazil

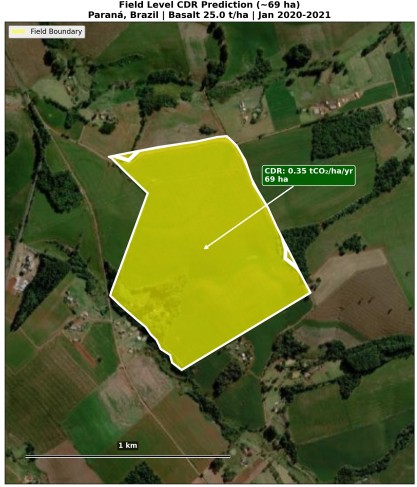

(b) Specific field in Paraná, Brazil

Figure 3: TerraNova in an operational setting: CDR rate predictions at national and field scales

## 4 EXPERIMENTS

### 4.1 QUANTITATIVE RESULTS

Table 1 summarizes our results. Note that FLAML determined XGBoost as its best model.

**Aurora representations transfer to weathering**: The consistent performance gap between Aurora-based and ERA5-based models (TerraNova: 0.8261 vs 0.3434; LSTM: 0.6131 vs 0.2660 on $R^2$) suggests that Aurora's learned representations encode atmospheric patterns relevant to geochemical processes that raw climate variables do not directly capture. This transfer from weather prediction to an entirely different domain: mineral dissolution kinetics, demonstrates that FMs can learn generalizable physical representations rather than task-specific features.

**Aurora representations improve upon tabular baselines:** FLAML's XGBoost achieves solid performance ($R^2 = 0.5792$, MAE $= 0.0914$) using only tabular site features without any weather information, indicating that soil, amendment, and spatial properties alone capture meaningful variation in CDR rates. However, TerraNova's combination of Aurora representations with multi-scale feature extraction improves both metrics ($R^2 = 0.8261$, MAE $= 0.0750$), suggesting that atmospheric dynamics captured by an FM provide complementary information that tabular approaches cannot access.

**Process-based models struggle without detailed calibration:** Without site-specific calibration, PHREEQC achieves near-zero $R^2$ (0.032), indicating that literature-derived parameterizations alone are insufficient for CDR rate prediction across diverse sites. This is expected: process-based models are designed to be calibrated against local observations, and our implementation used parameters directly from literature with limited tuning. Future work should compare against fully calibrated

process-based models to better characterize the relative strengths of each approach across different data availability scenarios.

**TerraNova in an Operational Setting:** Figure 3 shows how TerraNova can generate predictions of CDR rates at national (3a) and field (3b) scales. A standard amendment of basalt with standard mineralogy and RSA with an application rate of 25 tons/ha starting from January 12th, 2020 for one year is considered. Soil and dynamic site properties are obtained from the same sources as the training set (see Appendix). These examples are illustrative; with continued improvement and extensive validation, models like TerraNova have the potential to aid deployment planning and verification for ERW.

## 4.2 INTERPRETABILITY ANALYSIS

Interpretability is essential for building trust in deep learning models for scientific applications, as a common criticism is that neural networks can be "right for the wrong reasons." When applied to scientific domains, AI interpretability can not only help ground models in physical reality, but may also advance scientific understanding by identifying drivers of variation that mechanistic approaches have missed. In geochemistry, where process interactions remain incompletely understood, this can be particularly valuable.

To evaluate whether TerraNova learns scientifically meaningful relationships, we construct paired synthetic ERW examples that isolate individual variables. For each test, we hold all inputs constant except one factor (e.g., soil moisture), generate high and low variants, and verify that model predictions shift in the direction consistent with geochemical understanding. We evaluate 100 paired examples per category.

Table 2: Interpretability tests evaluating whether TerraNova predictions align with established geochemical intuition. Pass rate indicates the fraction of paired examples where predictions match expected directional effects.

| Factor | Scientific Basis & Expected Effect | Pass Rate |
|---|---|---|
| Soil Moisture | Water availability governs dissolution reactions and ion transport. Higher moisture → higher CDR rate. | 100/100 |
| Temperature | Warmer temperatures accelerate reaction kinetics. Higher temperature → higher CDR rate. | 55/100 |
| Reactive Surface Area (RSA) | RSA determines mineral-water contact; ERW amendments increase available surface area. Higher RSA → higher CDR rate. | 100/100 |
| Soil Texture | Sandy soils have higher permeability, promoting dissolution. Sandy > clay for CDR rate. | 74/100 |

Table 2 shows that TerraNova's predictions align with geochemical understanding for most factors, with perfect pass rates for soil moisture (100/100), RSA (100/100), and solid performance on soil texture (74/100).

However, temperature achieves only 55/100, a notable gap. While the Arrhenius relationship predicts faster reaction kinetics at higher temperatures, real-world weathering involves competing effects: warmer temperatures could be geographically confounded with highly weathered tropical soils, depleted cation pools, and distinct hydrological regimes. These correlations are encoded in both the learned weights and Aurora's latent representations. The model may be capturing these confounded relationships from the training distribution rather than isolating the pure temperature effect. This highlights a limitation of data-driven approaches: they learn correlational structure from observed data, which may not generalize when variables are artificially decoupled. Future work could explore causal representation learning or physics-informed constraints to better disentangle these factors.

## 5 DISCUSSION

### 5.1 TOWARDS AI-BASED MRV

These results demonstrate the promise of deep learning models to learn weathering dynamics without explicitly encoding the underlying geochemical mechanisms. Notably, Aurora-based models consistently outperform their ERA5 counterparts across architectures, suggesting that FMs capture latent atmospheric dynamics relevant to weathering that raw reanalysis variables do not represent.

The interpretability analysis provides evidence that TerraNova learns physically meaningful relationships, not merely exploiting dataset artifacts. High pass rates on soil moisture, RSA, and solid performance on soil texture indicate the model captures established scientific drivers. However, the temperature results reveal limitations: the model may learn confounded relationships from observational data rather than isolating causal mechanisms. This underscores that data-driven approaches should complement, rather than replace, mechanistic understanding.

While our quantitative results are primarily driven by natural weathering data, the interpretability analysis on synthetic ERW scenarios provides initial evidence that the learned dynamics may transfer, though extending to ERW in practice will require further extensive validation.

Together, these findings suggest a path for AI to aid ERW MRV and complement current ground-truth MRV. Current verification approaches require expensive field sampling and laboratory analysis, creating a bottleneck that limits deployment scale. With further training and extensive validation on data from global deployments, models like TerraNova have the potential to enable continuous, low-cost monitoring, which could accelerate the scaling of ERW to climate-relevant gigatons of CDR. More broadly, this work shows one pathway by which AI and FMs can help address climate change.

### 5.2 FUTURE WORK

We demonstrate the potential of FMs for CDR prediction. Several directions could extend this work and further improve models like TerraNova:

1. **Expand Training Data:** Our dataset has limited ERW field trials. Incorporating data from active ERW deployments, non-basaltic catchments, and controlled mesocosm experiments would improve coverage of the conditions encountered in practice.

2. **Validation on operational ERW sites:** While we validate on held-out chronosequences, true generalization requires evaluation on independent ERW deployments with ground-truth CDR measurements that could be obtained from ERW companies, research institutes, and efforts like the Cascade Data Quarry (Cascade Climate, 2025). This will be critical for enabling operational implementation.

3. **Richer Input Features:** Management practices (tillage, crop rotation, irrigation) influence weathering rates, but are sparsely represented in current data. Incorporating agronomic variables could improve predictions for agricultural ERW deployments.

4. **Extended Prediction Targets and Uncertainty Bounds:** Multi-task models predicting ERW's other effects like soil pH and crop yields would better support deployment decisions and improve interpretability. Additionally, including uncertainty bounds would improve confidence.

5. **More Comprehensive Comparison with Process-Based Models:** Our evaluation used a single process-based model with minimal calibration, representing a limited baseline. Future work should compare against fully calibrated implementations, potentially leveraging community efforts such as RockMIP (Leverhulme Centre for Climate Change Mitigation, 2024). Understanding the relative strengths of mechanistic and data-driven approaches across different calibration regimes is essential for developing robust MRV.

6. **Two-Stage Training:** In this work, we focused on training on natural weathering and ERW data together. We could also explore a two-stage training approach: pre-training on just natural weathering to learn general weathering dynamics and then fine-tuning on ERW data.

7. **Causal Interpretability:** Our synthetic perturbation tests reveal correlational patterns, but cannot establish causality. Techniques from causal representation learning could better disentangle confounded variables like temperature and moisture.

8. **Additional FMs:** We use Aurora, but comparison with other atmospheric FMs (e.g., GraphCast, Pangu-Weather) could reveal which representations best transfer to rock weathering.

9. **Extending FMs to Other Earth System Processes:** This work demonstrates that atmospheric FMs like Aurora can capture weathering-relevant dynamics beyond what raw climate variables provide. A natural extension is applying this approach to other Earth system processes. More broadly, the ability of FMs to transfer learned representations to unseen Earth system processes offers a compelling benchmark for evaluating their generality.

ACKNOWLEDGMENTS

We gratefully acknowledge Katya Larina and Jackson Harris for their invaluable contributions to dataset assembly and curation. We thank Prof. David Lobell, Prof. Scott Fendorf, and Prof. Ulrich Mayer for their expert review and domain feedback, which significantly strengthened this work. We thank the anonymous reviewers for their constructive feedback. We thank the Microsoft Aurora team for their generous support and responsiveness in helping us build with Aurora.

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

## A APPENDIX

### A.1 FEATURE VARIABLES

Table 3: Summary of input features and target variable used in TerraNova, organized by category. Note that ERA5 features are input into the frozen Aurora encoder. If a given variable isn't available from the original dataset (Hartmann et al., 2019; Agência Nacional de Águas e Saneamento Básico (ANA); Instituto Água e Terra, n.d.), it is obtained from an external source as shown below.

| Category | Variable | Units | External Source |
|---|---|---|---|
| ERA5 | *Surface:* | | |
| | 2m Temperature | K | ERA5 (Hersbach et al., 2020) |
| | 10m U-wind | m/s | ERA5 |
| | 10m V-wind | m/s | ERA5 |
| | Mean Sea Level Pressure | Pa | ERA5 |
| | *Static:* | | |
| | Land-Sea Mask | – | ERA5 |
| | Soil Type | – | ERA5 |
| | Surface Geopotential | $m^2/s^2$ | ERA5 |
| | *Atmospheric (13 levels):* | | |
| | Temperature | K | ERA5 |
| | U-wind | m/s | ERA5 |
| | V-wind | m/s | ERA5 |
| | Specific Humidity | kg/kg | ERA5 |
| | Geopotential | $m^2/s^2$ | ERA5 |
| Static Soil | Sand Content | % | SoilGrids (Poggio et al., 2021) |
| | Silt Content | % | SoilGrids |
| | Clay Content | % | SoilGrids |
| Dynamic Site | Soil pH | pH | SoilGrids |
| | Soil Cation Exchange Capacity | cmol/kg | SoilGrids |
| | Soil Moisture | mm | GLDAS (Rodell et al., 2004) |
| | Soil Runoff | mm/day | UNH/GRDC (Fekete et al., 2002) |
| | Evapotranspiration | mm/day | Willmott & Matsuura (2001) |
| | Soil Organic Carbon | g/kg | SoilGrids |
| | Soil Erosion | kg/ha/day | GLOSEM (Borrelli et al., 2017) |
| | Soil Porosity | % | GLDAS |
| | Base Saturation | % | HWSD (FAO & IIASA, 2023) |
| | Net Primary Productivity | $gC/m^2$/day | MODIS (Zhao et al., 2005) |
| | Precipitation | mm/year | ERA5 |
| Amendment | CaO | % | – |
| | MgO | % | – |
| | $Na_2O$ | % | – |
| | $K_2O$ | % | – |
| | MnO | % | – |
| | $SO_3$ | % | – |
| | $P_2O_5$ | % | – |
| | Quartz | % | – |
| | Calcite | % | – |
| | Actinolite | % | – |
| | Piemontite | % | – |
| | Chlorite | % | – |
| | Titanite | % | – |
| | Muscovite | % | – |

*Continued on next page*

*Table 3 continued from previous page*

| Category | Variable | Units | External Source |
|---|---|---|---|
| | Albite | % | – |
| | Reactive Surface Area | $m^2$/g | Calculated if unavailable |
| | Application Rate | t/ha | Calculated if unavailable |
| Spatial | Latitude | ° | – |
| | Longitude | ° | – |
| | Field Area | ha | – |
| Temporal | Date | Day, Month, Year | – |
| Target | CDR Rate | t $CO_2$e/ha/yr | Calculated if unavailable |

