# OpenReview forum: "From Weather to Weathering: Foundation Models for Carbon Removal via Rock Weathering"
_ICLR.cc/2026/Workshop/FM4Science — ICLR 2026 Workshop FM4Science Poster_

### Official Review · Reviewer_3ooL · 2026-02-15
**This manuscript introduces TerraNova, the first deep-learning model to leverage latent representations from an atmospheric foundation model (Aurora) to predict Carbon Dioxide Removal (CDR) rates from Enhanced Rock Weathering (ERW)**

**Rating:** 9
**Confidence:** 4

**Review:**

This is another good work on foundation model. The work provides robust methodology for transforming sparse geochemical measurements into a supervised learning task. The authors assembled a massive dataset from the GloRiCh and ANA/IAET databases, focusing on basaltic catchments, and paired measurements at different times to create chronosequences with varying temporal gaps

Strengths:
* Novel Cross-Domain Transfer: Proves that latent representations from an atmospheric FM (Aurora) contain essential physical signals for geochemistry that raw ERA5 variables do not.

* Massive Dataset Construction: Utilizes 193,401 chronosequences, providing the most comprehensive data-driven look at rock weathering to date.

* Multi-Scale Feature Extraction: Captures local (field-level), regional, and global climate context, allowing for precise field-level predictions (as seen in the Paraná, Brazil operational example).

* Superior to Process Models: Outperforms literature-derived PHREEQC models, which often fail without labor-intensive site-specific calibration.

* Strong Interpretability Alignment: Passes 100/100 tests for soil moisture and reactive surface area (RSA) effects, grounding the AI in geochemical reality


Weakness:
* Causal Confounds: The model struggled with temperature effects (55/100 pass rate), likely due to high correlations between temperature and soil age in the training data, highlighting the danger of correlational learning in physics.

* Lithological Bias: The model is currently restricted to basaltic feedstock; its performance on other silicate minerals (e.g., wollastonite or olivine) is unknown.

* Limited ERW Data: The training set is dominated by natural weathering; while the physics are shared, the transition from catchment-scale to plot-scale management might introduce unseen variables

---

### Official Review · Reviewer_7W7T · 2026-02-21
**Creative foundation model transfer with architectural drawbacks**

**Rating:** 6
**Confidence:** 5

**Review:**

This paper introduces a deep learning model, TerraNova, to predicts CO2 removal rates for ERW. The paper overcomes ERW data scarcity by constructing a large chronosequence dataset from natural basaltic catchments.

Applying a weather foundation model to geochemical kinetics is a very original approach. TerraNova delivers strong performance and outperforms baselines that rely solely on tabular features or raw climate data.

The experimental evaluation contains notable flaws. Comparing a heavily trained neural network against an uncalibrated process-based baseline (PHREEQC) artificially inflates the performance gap.

While the core concept is strong, the specific network architecture contains inefficiencies and design flaws. First, the model severely over-parameterizes low-dimensional inputs. Projecting a 3-dimensional vector (such as static soil properties or spatial coordinates) into a 512-dimensional hidden space is massive overkill. This design choice likely introduces dead neurons and wastes memory capacity.

The architecture employs a strictly "late fusion" strategy. The five distinct tabular categories are processed in complete isolation through separate MLPs. They are only concatenated at the very end right before the final prediction head. In real-world geochemistry, properties like soil pH and rock mineralogy are highly coupled. By keeping these feature streams isolated, the network loses the opportunity to learn critical early cross-domain interactions. I suggest it might be better to concatenate the tabular features early and pass them through a single shared MLP.

Finally, Figure 2 is poorly constructed. Stacking five identical MLP blocks vertically makes a fundamentally straightforward architecture appear unnecessarily complex. The notation for the decoder is also somewhat non-standard and could be simplified for better readability.

---

### Meta-Review · Area_Chair_CTVG · 2026-02-27

**Recommendation:** Accept (Poster)
**Confidence:** 3

**Metareview:**

This submission has received one very positive review with a "strong accept" and a "marginally above acceptance threshold" review.

After reading the reviews, I recommend this paper for "acceptance" and strongly suggest that the authors of this submission consider the suggestions of reviewer "7W7T".

---

### Decision · Program_Chairs · 2026-03-03

Accept (Poster)